# n-Heptadecane-Impregnated Wood as a Potential Material for Energy-Saving Buildings

**Ahmet Can [1,2,\*]** and **Jure Žigon [3]**

1   Faculty of Forestry, Bartın University, 74100 Bartın, Turkey
2   Forest Industry Engineering, Bursa Technical University, 16310 Bursa, Turkey
3   Department of Wood Science and Technology, Biotechnical Faculty, University of Ljubljana, Jamnikarjeva 101, 1000 Ljubljana, Slovenia
*   Correspondence: ahmet.can@btu.edu.tr or acan@bartin.edu.tr; Tel.: +90-5379547177

**Abstract:** Phase change materials (PCMs) are ideal for thermal management solutions in buildings. This is because they store and release thermal energy during melting and freezing. Spruce (*Picea orientalis* (L.) Peterm.) sapwood was impregnated with n-heptadecane (100%) as a PCM. The decay-resistance properties and thermal energy storage (TES) characteristics of the n-heptadecane-impregnated wood were studied. The phase change properties of n-heptadecane (nHD)-impregnated wood were characterized by Fourier-transform infrared (FTIR), thermogravimetry (TGA), differential scanning calorimetry (DSC) and X-ray diffraction (XRD) analyses. As confirmed by DSC analysis, nHD-impregnated wood demonstrated moderate performance in storing and releasing heat during the phase change process. Significant increases were observed in the 2800–3000 cm$^{-1}$ and 1471 cm$^{-1}$ peaks in FTIR spectra of wood samples impregnated with nHD, which showed C–H stretching in methyl and methylene groups and asymmetric deformation vibration of the paraffin methyl group (CH$_3$–) and C–O stretch in lignin, respectively. It was observed that there was a change in the crystal structure of spruce wood samples after nHD impregnation. This study revealed that PCMs are resistant to wood-destroying fungi. The performance of nHD-impregnated spruce wood proves that it can be used as a thermal regulating building material to reduce energy consumption. In addition, it has been proven on a laboratory scale that the PCM used is highly resistant to biological attacks. However, large-scale pilot studies are still needed.

**Keywords:** n-heptadecane; phase change material; spruce wood; wood impregnation

## 1. Introduction

In recent years, energy demand has increased due to economic crises, population growth, and wars, and it has become important once again that each country should gain its own energy independence. Countries are working to increase energy production from renewable resources for this need. However, renewable energy resources such as wind, solar, geothermal, and hydroelectricity are not enough to meet the total energy demand of countries and will not be enough in the future. In recent years, researchers have been increasing their efforts to grow the use of renewable energy and to develop environmentally friendly materials [1–3]. Thermal energy storage (TES) can somehow meet the increasing need for energy conservation [4–7].

Phase change materials (PCMs) used as TES have high heat storage capabilities, reliability, practicability, and moderate variation in volumes and temperatures [7–9]. Thermophysical, kinetic (the PCMs must have minimum subcooling and sufficient crystallization rate to form crystals), chemical properties (long lifetime, building compatibility, nonflammable, noncorrosive, atoxic) and economy are the key factors in selecting suitable PCMs for building applications. An efficient PCM should have high latent heat and high thermal conductivity [10]. In addition, a suitable PCM should have a melting temperature

above 20 °C to meet a comfortable indoor temperature. According to the American Society of Heating, Refrigerating and Air Conditioning Engineers (ASHRAE), there should be a certain temperature ranges in summer and winter periods. According to their recommendations, comfortable room temperature is 23.5–25.5 °C in summer and 21.0–23.0 °C in winter [11]. PCMs must have a high density that allows more heat to be stored and physical properties that cause small volume changes [12]. There are essentially two different types of PCMs. These are PCMs with liquid–solid and solid–solid behavior, which are capable of absorbing and releasing large amounts of latent heat during phase change [9]. Paraffin is one of the organic PCMs (OPCMs), and contains straight-chain alkanes (n-alkanes). Paraffin waxes are used as PCMs due to some of their superior properties, such as melting and solidifying at wide temperature ranges, low vapor pressure, large latent heat capacity (near 200 J/g), good thermal and chemical stability, negligible supercooling, and low melting point. Because of this properties, n-heptadecane (nHD) can be used in smart textiles and thermoregulated fibers [9].

In addition to these superior properties of PCMs, leaching to the environment is the most important disadvantage [13,14]. To overcome this disadvantage, PCMs are either microencapsulated in polymer shells [15] or incorporated in a porous supporting material [16]. When encapsulated in the shell, the PCMs can melt. The shell material prevents leakage [17], the negative impact of the external environment on the PCMs is prevented and the life of the PCM is extended [18]. Moreover, the encapsulation of the PCMs increases the heat transfer area; thus, since the heat transfer area of the latent heat storage system is increased, the transfer rate also increases [19]. PCMs are easier to incorporate into building materials, because the use of paraffin in the microencapsulation method is easier to manufacture, cheaper, sustainable, and more stable [20]. Various materials containing PCMs are used, such as attapulgite, diatomite, expanded graphite, expanded vermiculite, and biochar. These materials, thanks to their porous structure, prevent the leaching of the material during the phase change to some extent [21–24]. Amini et al. [25] prepared composite PCMs by using Scotch pine (*Pinus sylvestris* L.) sapwood as support material and different concentrations of capric acid (CA) (20%, 40%, 60%, and 80%) as PCMs.

In recent years, the use of wood in the construction of high-rise buildings has been increasing due to its high strength-to-weight ratio, good insulation properties, and renewability and sustainability [26–28]. However, in addition to the superior properties of wood material, some negative features are present, such as dimensional instability and susceptibility to decay due to climate variations (especially in outdoor conditions) and poor adhesion due to its high hydroscopic nature [29–31]. It is important to improve the functionality of the wood used in high-rise buildings. The increment in heat storage capacity of wood impregnated with phase change agents is an example of such approach.

In the current study, spruce (*Picea orientalis* (L.) Peterm.) sapwood was successfully impregnated with nHD, in order to improve the energy storage property of wood. On good grounds, nHD-impregnated wood (PCMW) was characterized chemically and thermally. The nHD and PCMW were examined using Fourier-transform infrared (FTIR), differential scanning calorimetry (DSC), X-ray diffraction (XRD) and thermogravimetry (TGA) analytical techniques. The nHD was chosen as PCM due to favorable phase change temperature (about 20 °C) and high latent heat (217 J/g) for passive thermal regulation of buildings. The present research aimed to (i) study the influences of n-HD on the thermal storage properties of wood, and (ii) study the performance of nHD-impregnated wood against wood-destroying fungi.

## 2. Materials and Methods

### 2.1. Materials

Spruce sapwood samples with dimensions of 5 × 15 × 30 mm were used for the decay test: six replicates for impregnated samples and six control samples. nHD (Merck Company, New York, NY, USA, ABD) was selected as PCM. nHD with the linear formula

$CH_3(CH_2)_{15}CH_3$, molecular weight of 240.48 g/mol and purity of $\geq$99.0% has a melting temperature range from 22 °C and a density of 0.78 g/mL at 25 °C.

### 2.2. Impregnation of Wood Samples with PCM (PCMW)

First, wood samples were dried to approximately absolute dry state. The impregnation of wood samples was carried out with nHD at 100% concentration. The impregnation of nHD was carried out in a vacuum chamber under a pressure of 850 mbar at 30 °C for 1 h and left at atmospheric pressure for 24 h. The weight percentage gain (WPG, %) for nHD-impregnated wood was 18.97%. After impregnation, the samples were conditioned in the climate chamber for 3 weeks at 23 °C and 65% relative humidity, and the wood mass was recorded prior to decay test.

### 2.3. Decay Test

The decay test was performed according to the principles of Bravery [32]. *Coniophora puteana* (Mad-515) as a brown rot fungus, and *Trametes versicolor* (L.) Lloyd (Mad-697) as a white rot fungus were cultured to a sterile malt extract agar in petri dishes. Wood samples were sterilized at 120 °C for 30 min before inoculating. After the inoculation, wood samples were incubated at 20 °C and 70% relative humidity for 12 weeks. The weight loss was represented on the basis of the oven-dry weight before and after the decay test.

### 2.4. Characterization of PCM and PCMW

The FTIR spectra of microcapsules and spruce wood were enlisted between 4000 and 500 cm$^{-1}$ wavelengths with 4 cm$^{-1}$ resolution using an ATR FTIR spectrometer (Bruker Optics GmbH, Ettlingen, Germany).

DSC analysis was done under a sustained stream of argon, with a cooling and heating rate of 5 °C/min. The test temperature range was 10–40 °C.

The thermal stability was done using a TG analyzer (Perkin Elmer Pyris™ 1 TGA) at a scanning rate of 10 °C min$^{-1}$ in the temperature range of 20–600 °C under a nitrogen atmosphere. All charts were smoothed and baseline-corrected using OPUS software (Bruker).

Phase content analyses of the samples were performed by X-ray diffraction (XRD, Bruker AXS/Discovery D8), using monochromatic Cu-K$\alpha$ radiation (l = 1.5406 Å).

## 3. Results and Discussion

FTIR spectroscopy has been widely used to confirm whether the modification process has taken place successfully [33,34]. The FTIR spectra of nHD and the PCMW are shown in Figure 1. In the spectrum of pure n-HD (Figure 1 (PCM)), the peaks at 2957 cm$^{-1}$, 2920 cm$^{-1}$, and 2852 cm$^{-1}$ conform with the symmetrical C–H stretching vibration of single-bond $CH_3$ and single-bond $CH_2$ groups, respectively. The peaks observed at 1363 cm$^{-1}$ and 1466 cm$^{-1}$ represent C–H deformation vibration bands [33,34].

FTIR spectra of spruce wood are given in Figure 1. These spectra (700–4000 cm$^{-1}$) simply show the spectra of wood: the 3342 cm$^{-1}$ peak shows strong main OH stretching, and 2800–3000 cm$^{-1}$ shows C–H stretching in methyl and methylene groups. The other characteristic peak at 1735 cm$^{-1}$ (acetyl groups in xylan and other nonconjugated carbonyls), C=O stretch in xylan (hemicelluloses) and the peaks at 1508 cm$^{-1}$ and 1427 cm$^{-1}$ show the aromatic skeletal vibration of lignin. The peaks at 1369 cm$^{-1}$ and 897 cm$^{-1}$ indicate the peaks of C-H deformation found in carbohydrates. The peak at 1265 cm$^{-1}$ indicates C–O stretch lignin ring in guaiacyl units and 1026 cm$^{-1}$ C–O vibration in cellulose and hemicelluloses [33,34].

Some changes in the chemical structure occurred with the impregnation of spruce wood samples with PCM. In general, the chemical structure of PCMs is also seen in wood samples. In particular, peaks at 2852 cm$^{-1}$, 2920 cm$^{-1}$, 2957 cm$^{-1}$ were also observed in wood samples. In addition, significant increases in the 1471 cm$^{-1}$ and 1265 cm$^{-1}$ peaks in the PCMW samples indicate asymmetric deformation vibration of the paraffin methyl group ($CH_3$–) and C–O stretch in lignin ring of guaiacyl units, respectively. In the

PCMW samples, a significant increase was observed at 1471 cm$^{-1}$, which indicates C–H deformation in lignin and carbohydrates. In addition, a 720 cm$^{-1}$ peak, which is one of the PCM characteristic peaks, appeared at 715 cm$^{-1}$ in wood samples.

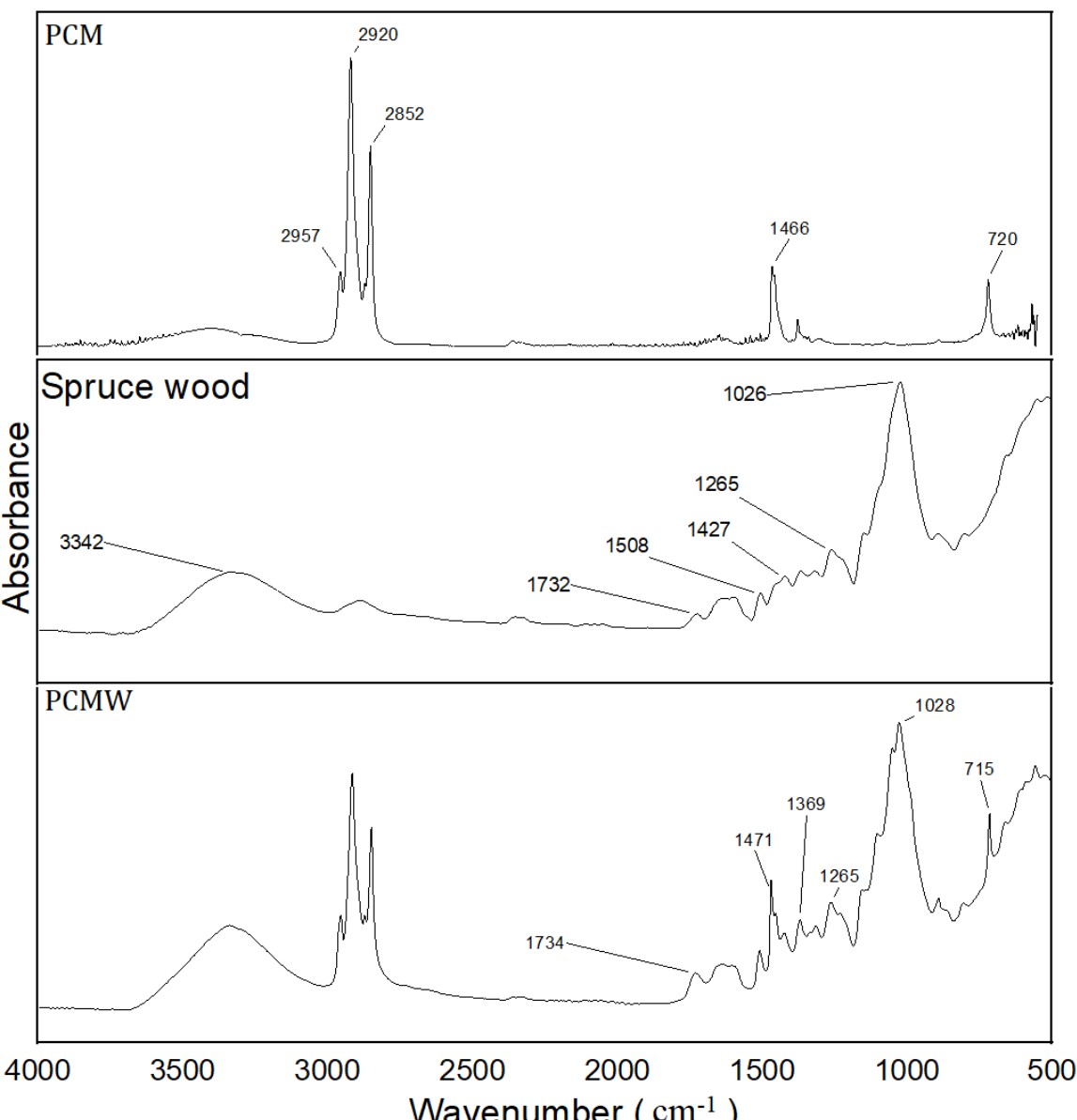

**Figure 1.** FTIR-ATR spectra of the PCM (nHD) and PCMW.

The DSC results are shown in Figures 2 and 3. Natural wood does not have melting or solidification temperature peaks. However, melting and solidification temperature peaks occurred in wood samples impregnated with nHD. The melting temperature of PCM is 22.07 °C, while it is determined as 19.42 °C for PCMW. The PCM-impregnated wood has the lower melting temperature at 19.42 °C than PCM with 22.07 °C. There is no significant difference between the two samples. This situation may have been caused by the (1) confinement of PCMs in the pores of the wood after the impregnation process, (2) the hydrogen bonding between the paraffin and the lignin, cellulose, and hemicellulose components of the wood, (3) and the effect of capillary forces between the PCM molecules and cell walls of the wood. The melting latent heat of the PCM is 168.71 J/g, while PCM-impregnated wood has lower melting latent heat (2.8127 J/g) than PCMs. nHD used as PCM has a long chain structure, and when used alone, it has an enthalpy of 168.71 J/g

at 22.07 °C. The enthalpy value decreased after the wood was impregnated with nHD. This shows that it has no effect on the enthalpy value of wood. In addition, with the nHD impregnation process, which has a long chain structure, the chains were broken and the enthalpy value decreased. From the DSC data of Figures 2 and 3, it can be seen that the solidifying temperature of PCM is 19.27 °C, which is higher than solidifying temperature of PCMW (17.43 °C).

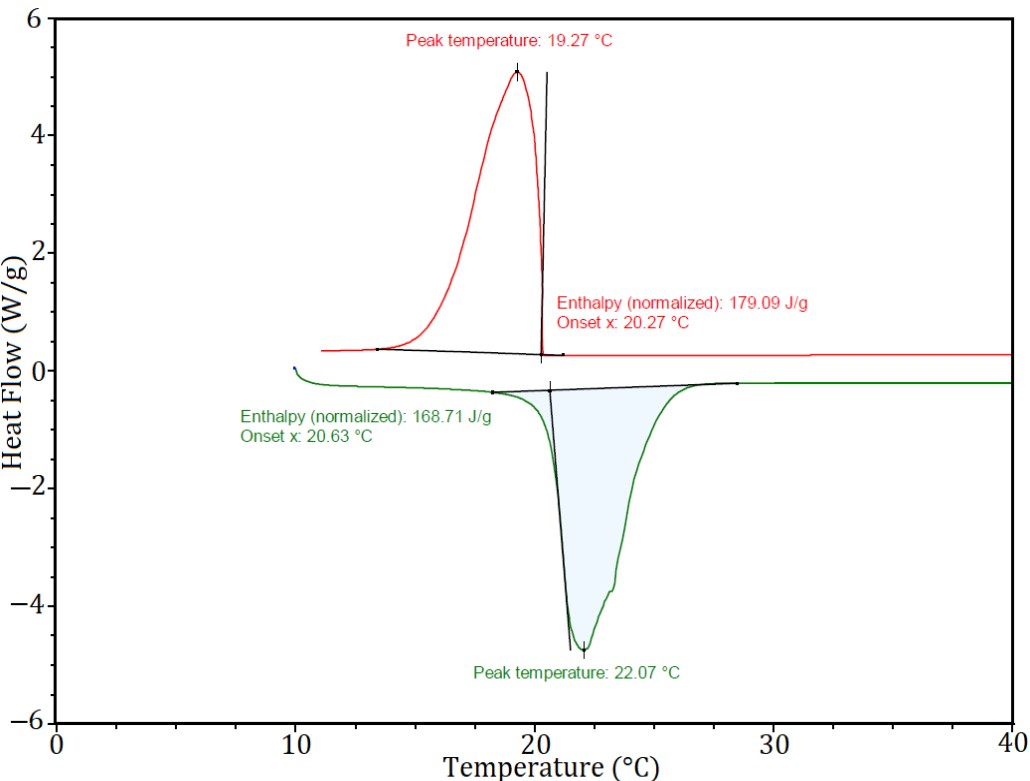

**Figure 2.** Melting (green) and solidifying (red) DSC curves of the PCM (nHD).

Figures 4 and 5 present the TGA thermograms of n-HD (PCM), spruce wood, and PCM-impregnated wood. The n-HD decomposition occurred in a single step between 90 °C and 216 °C, without remaining residue. The spruce wood samples exhibited degradation in two steps (Figure 5). The first weight loss stage was attributed to the degradation of hemicelluloses, which usually occurs between 100 °C and 365 °C, while the second mass loss corresponded to the cellulose decomposition at a slightly higher temperature range (between 270 °C and 400 °C). Spruce wood lost about 1% of its weight at 100 °C, which may be attributed to the loss of physically adsorbed moisture and volatile compounds. Between 100 °C and 600 °C, wood lost 82.41% of its weight and the residual weight was 17.40% at the end. The maximum weight loss of spruce wood occurred at 356 °C. The most rapid weight loss happened at temperature of 365 °C for PCM-impregnated wood with a rate of 81.50%. The PCM-impregnated wood left lower residue of 17.35% than normal wood. This result indicates that PCM-impregnated wood had thermal stability at extreme temperatures compared to the PCM. In PCMW samples, degradation took place in two stages: nHD in the first stage and wood components in the second stage. It is thought that nHD chains are broken in the first stage of degradation, while cellulose, one of the wood components, is broken down in the second stage.

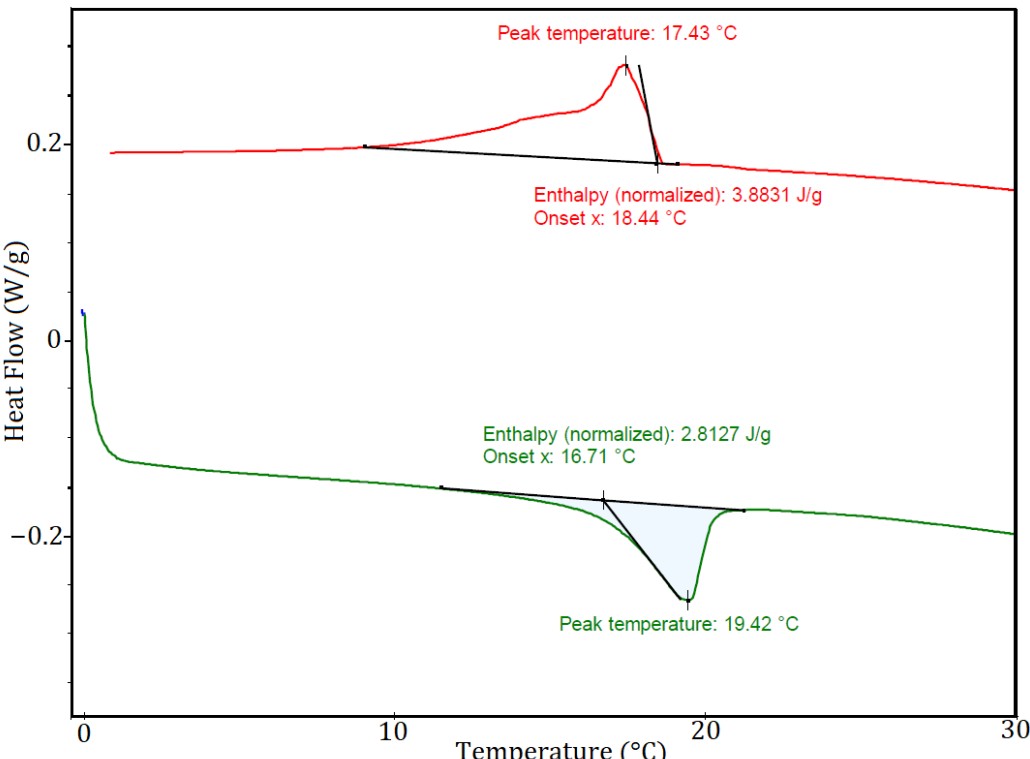

**Figure 3.** Melting (green) and solidifying (red) DSC curves of the PCM-modified wood (PCMW).

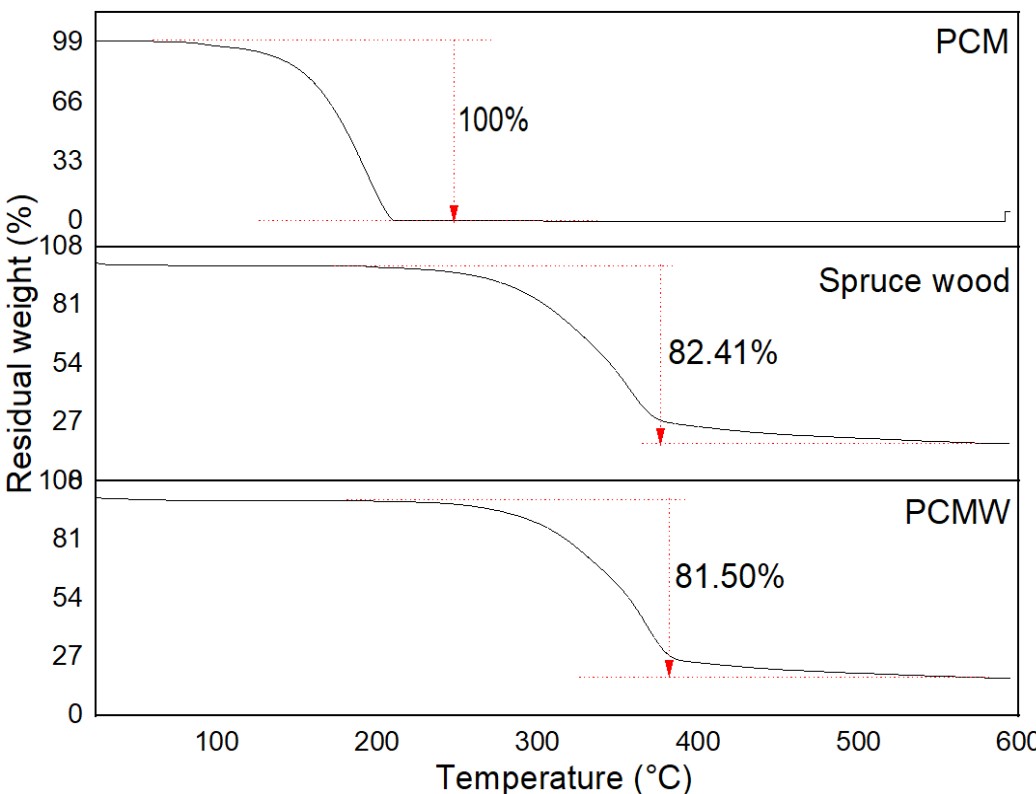

**Figure 4.** Residual weight of the PCM (n-HD)- and PCM-impregnated wood (PCMW).

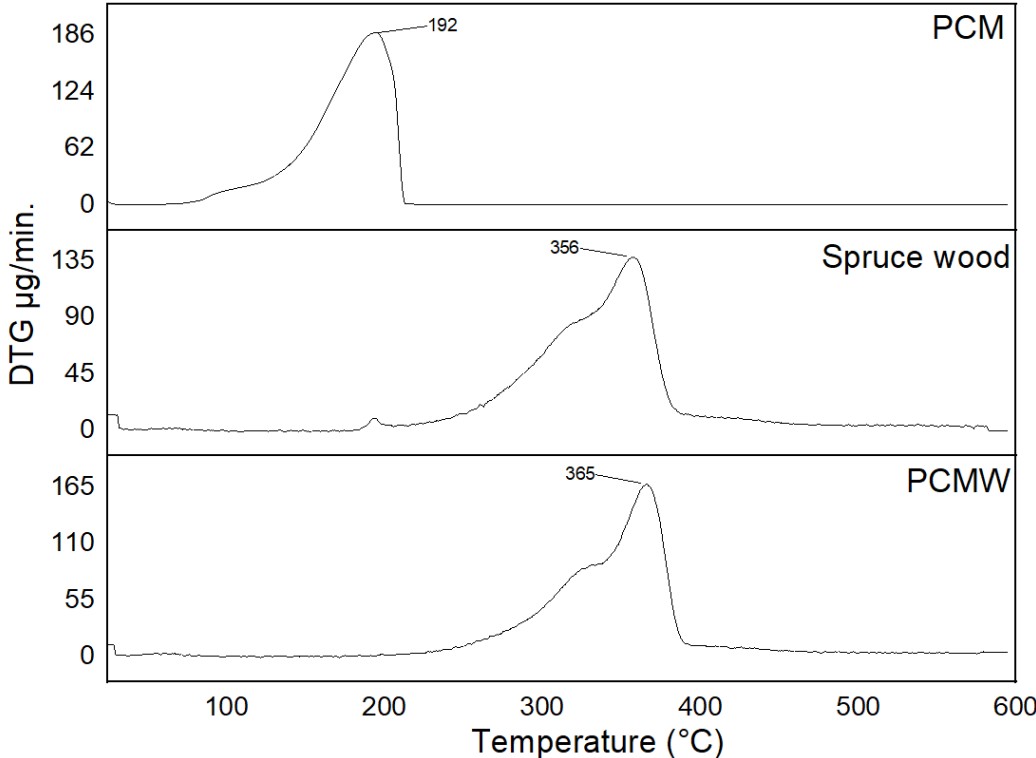

**Figure 5.** DTG of the PCM (n-HD)- and PCM-impregnated wood (PCMW).

The XRD diffraction graph of spruce wood and PCMW is given in Figure 6. The diffraction peaks of spruce wood were found at 17.0°, 21.84°, 24.64°, 33.98° and 43.98°. These characteristic peaks represent the crystal structure of cellulose in wood samples. The peaks obtained in 33.98° and 43.98° differ from the literature [35]. This may be due to differences in the type of spruce. It is observed that the crystal structure in wood samples impregnated with nHD changed. Moreover, the characteristic peaks at 2θ value of 23.67° and 33.98° denote the crystal structure of wood, while 13.77° and 16.50° denote the crystal structure of nHD.

The highest mass losses of 22.80% and 22.56% eventuated when the spruce sapwood was subjected to white rot fungus and brown rot fungus for 12 weeks' incubation, respectively. As a result of the decay test, it can be stated that spruce wood had high weight losses, and this situation is in the "nondurable" class according to the EN 113 standard [36]. In addition, for the test to be considered successful, at least 20% weight loss is expected in the control samples.

Although high weight loss occurred in control samples, less than 3% of weight loss was obtained in PCMW (test samples). The PCMW were usually more intensively attacked by the white rot fungus (from 1% to 10%) than by the brown rot fungus (from 1% to 8%). The image of the samples after the decay test is given in Figure 7.

The studies on paraffin-based materials report that wood attacked by wood-rotting fungi may degrade more slowly when impregnated with waxes. Filling the wood cell walls and cell lumens with paraffin wax prevents the diffusion of fungal enzymes from going deeper. Thus, the diffusion between the hyphae of decaying fungi in wood, and wood structural components is slowed down. Specific changes in the molecular structure of wood that cause an increase in its resistance to decay occur as a result of parallel PCM impregnation. Inhibition of fungal growth here also results from polysaccharide dihydroxylation, resulting in lower moisture content of the wood [37–39]. In this study, high resistance was achieved against *T. versicolor* and *C. puteana* fungi with nHD. In terms of the suitability of the use of phase change materials in buildings, studies should be conducted on resistance to soft rot fungi, mold fungi, insects and termites.

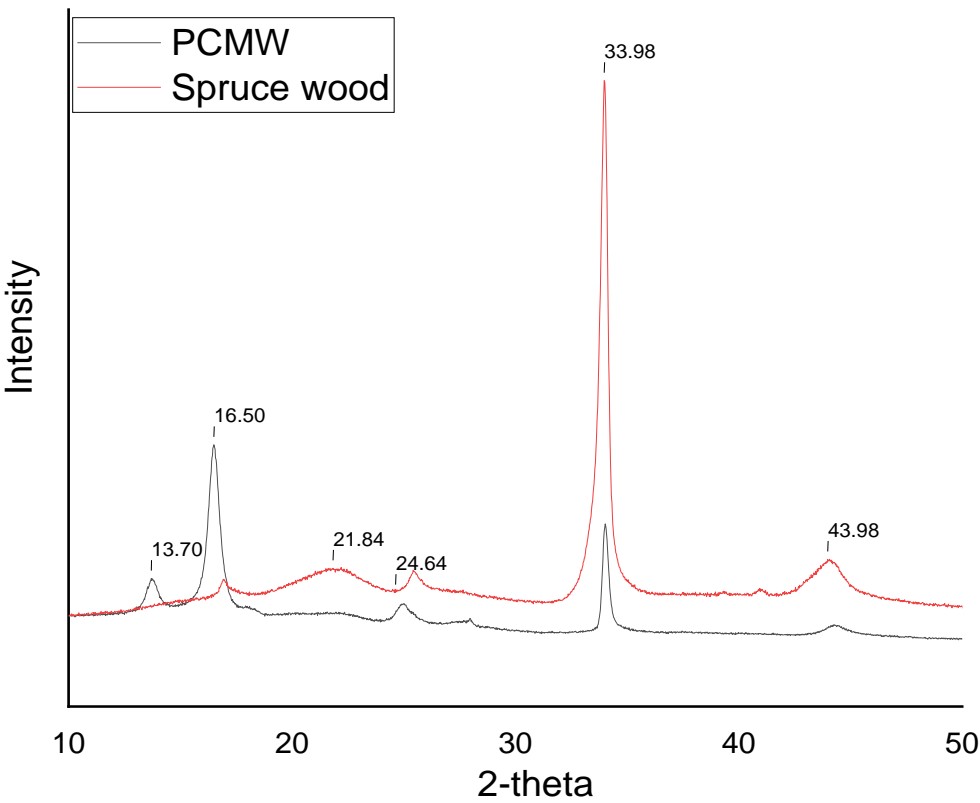

**Figure 6.** XRD patterns of spruce wood and PCMW.

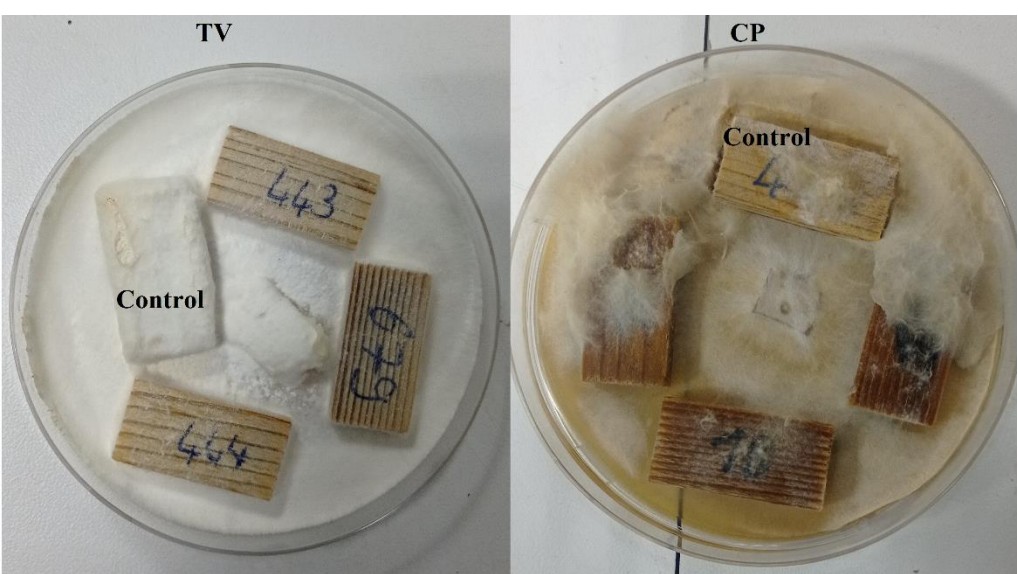

**Figure 7.** Samples after decay test: white rot fungus (**left**) and brown rot fungus (**right**).

## 4. Conclusions

In this study, important characteristics of spruce sapwood impregnated with nHD were revealed. Wood samples were impregnated with 100% concentrated nHD in a vacuum process, in order to fabricate a smart biocomposite for wooden building applications. The thermal and decay testing of PCMW, envisaged as being used as a thermal regulating component in buildings, showed positive results. The DSC results of the nHD-impregnated wood signified good energy storage/release capacity with appropriate phase change temperatures for building applications. The TG/DTA measurements revealed that nHD-impregnated wood left a higher residue of 18.50% at 600 °C than that of the

untreated wood with 11.59%. The main findings of the decay test were that the tested nHD-impregnated wood was durable against *Coniophora puteana* (brown rot), and *Trametes versicolor* (L.) Lloyd (white rot). We suggest PCMW as an energy-saving building material to ensure indoor temperature quality.

**Author Contributions:** Conceptualization, A.C; methodology, A.C. and J.Ž.; software, A.C.; validation, A.C. and J.Ž.; investigation, A.C.; resources, A.C.; data curation, A.C.; writing—original draft preparation, A.C.; writing—review and editing, J.Ž.; visualization, A.C.; project administration, J.Ž.; funding acquisition, J.Ž. All authors have read and agreed to the published version of the manuscript.

**Funding:** J.Ž. acknowledges the financial support from the Slovenian Research Agency (research program funding No. P4-0430, "Forest-wood value chain and climate change: transition to circular bioeconomy").

**Data Availability Statement:** No data were used for the research described in the article.

**Acknowledgments:** We would like to thank Bursa Technical University, Faculty of Forestry Dean's Office for the use of the FTIR device.

**Conflicts of Interest:** The authors declare no conflict of interest.

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
