# Peer review of "n-Heptadecane-Impregnated Wood as a Potential Material for Energy-Saving Buildings"

_forests, doi:10.3390/f13122137_

Round 1

Reviewer 1 Report

Overall this manuscript is written well, however, If the author adds more results on energy storage will be preferred. this is because this paper aims at energy storage. DSC and TGA are not sufficient

Author Response

Dear Reviewer,

Thank you for your valuable comments and suggestions on our manuscript.

Please fint attached our answers and the revised version of the manuscript.

Best regrads.

Reviewer 2 Report

1. The resolution of Figure 1 needs to be further improved and I think that a comparison of the three samples in a single figure would give a more intuitive view of the changes in functional groups.

2、The characterization test of X-ray diffraction can be implemented to study the crystallization of crystals before and after modification.

3. The authors have not designed the experiments in such a way that the microstructure of the samples can be analyzed. I believe that scanning electron microscopy tests are necessary, and that SEM can be used to microscopically observe the morphology and structure of the samples before and after impregnation and before and after fungal infestation, which would further corroborate the authors' statements.

4. The design of the experiments is lacking, and the required characterization and tests are missing to corroborate the results obtained.

Author Response

Dear Reviewer,

Thank you for your valuable comments and suggestions on our manuscript.

Please find attached our answers and the revised version of the manuscript. 

Best regrads.

Reviewer 3 Report

Please see the attachment. Thanks.

Author Response

(The authors gave the same response as above.)

Round 2

Reviewer 2 Report

I agree with the scientific publication.